# CIM-Based Smart Pose Detection Sensors

**DOI:** 10.3390/s22093491

**Published:** 2022-05-04

**Authors:** Jyun-Jhe Chou, Ting-Wei Chang, Xin-You Liu, Tsung-Yen Wu, Yu-Kai Chen, Ying-Tuan Hsu, Chih-Wei Chen, Tsung-Te Liu, Chi-Sheng Shih

**Affiliations:** 1Department of Computer Science and Information Engineering, National Taiwan University, Taipei 10617, Taiwan; billqwer1687@gmail.com (T.-W.C.); r10944004@csie.ntu.edu.tw (X.-Y.L.); 2Department of Electrical Engineering, National Taiwan University, Taipei 10617, Taiwan; r08943051@ntu.edu.tw (T.-Y.W.); yukai030405@gmail.com (Y.-K.C.); ythsu@eecs.ee.ntu.edu.tw (Y.-T.H.); ttliu@ntu.edu.tw (T.-T.L.); 3Center of High Performance and Scientific Computing Technology, National Taiwan University, Taipei 10617, Taiwan; piny.chen@mediatek.com; 4MediaTek Inc., Hsinchu 30078, Taiwan; 5Graduate Institute of Networking and Multimedia, National Taiwan University, Taipei 10617, Taiwan

**Keywords:** analogy computing, smart sensors, non-ideality errors

## Abstract

The majority of digital sensors rely on von Neumann architecture microprocessors to process sampled data. When the sampled data require complex computation for 24×7, the processing element will a consume significant amount of energy and computation resources. Several new sensing algorithms use deep neural network algorithms and consume even more computation resources. High resource consumption prevents such systems for 24×7 deployment although they can deliver impressive results. This work adopts a Computing-In-Memory (CIM) device, which integrates a storage and analog processing unit to eliminate data movement, to process sampled data. This work designs and evaluates the CIM-based sensing framework for human pose recognition. The framework consists of uncertainty-aware training, activation function design, and CIM error model collection. The evaluation results show that the framework can improve the detection accuracy of three poses classification on CIM devices using binary weights from 33.3% to 91.5% while that on ideal CIM is 92.1%. Although on digital systems the accuracy is 98.7% with binary weight and 99.5% with floating weight, the energy consumption of executing 1 convolution layer on a CIM device is only 30,000 to 50,000 times less than the digital sensing system. Such a design can significantly reduce power consumption and enables battery-powered always-on sensors.

## 1. Introduction

Using connected and intelligent sensors in the indoor spaces to collect data for 24×7 is desired to avoid uncomfortableness and lost tracking due to forgettable memory. For example, observing the gait velocity, activities, and safety at long-term care center or private space for the elderly or disabilities [1,2]. However, the sensing and computation will consume significant amount of resources, including energy, computation, storage, and network bandwidth for long-term monitoring. The above pitfalls limit the use of deploying connected sensors in indoor spaces.

One example is using low-resolution image sensors to evaluate the quality of sleep without wearable sensors and revealing privacy by counting the number of turnovers. The users do not need to wear any devices, and no wires are attached to the bed. Figure 1 shows the examples of color and corresponding thermal images of the users: Figure 1a,b show the images for laying down and turning to the right.

The example shows that the difference of thermal images could be used to identify the pose of the users without revealing their identifies. However, the process of identifying turn-overs requires increasing the resolution of the thermal images and executing the algorithm to detect the turn-overs using a sequence of images. The algorithm can be executed either on the sensors or on a remote server. Unfortunately, both deployments consume non-negligible amount of energy, and it is not feasible for mass deployment and long-term monitoring.

This work designs and evaluates the framework of using CIM processing elements, rather than von Neumann architecture micro-processors, to process thermal images for pose detection. The developed framework can significantly reduce the resource use, including energy consumption and storage size, without lowering the detection accuracy. Conducting a convolution layer whose input size is 30×40 pixels and kernel size is 7×7 consumes 14.55 mJ on a Raspberry PI 3 but only consumes 0.00029 mJ on CIM, which reduces the energy consumption 50,000 times.

CIM is an emerging technology which can do matrix-vector multiplies at 25 Mhz or higher [3,4,5,6] with low-energy consumption. It can be used to enhance the resource use of conducting neural network based algorithms. CIM chips usually consist of three parts. The first part is the random access memory (RAM), which is used to store the weight of the neural network model. In 2017, Zhang [3] reported the use of the 6T SRAM and amplifiers to conduct an AdaBoost classifier. In 2021, Chen [4] reported to use the DRAM as the memory part of CIM. Unlike SRAM, which can only store zero or one, DRAM can store analog weights. The second part is the DAC (Digital-to-Analog Converter), which transforms the digital inputs to analog voltage or a pulse. Chen’s work [4] shows that one can store the weights on DRAM, transform the input to a 30 ps to 750 ps pulse and send to the read-enable pin. The voltage of the Read-Line (RL) is proportional to the dot product of inputs and weights. The last part is an ADC (Analog-to-Digital Converter), which converts the analog voltage results back to the digital data. This work uses the CIM chip designed by Liu and others [6] as the processing unit to execute neural network algorithms. This chip uses an 128×64 CIM SRAM, which consists of 7-bit DACs and 7-bit ADCs. It also designs a sign-bit on input DAC, which can switch the precharge RL and makes the CIM capable of supporting negative inputs.

Figure 2 illustrates the difference between the von Neumann and CIM architecture for pose detection. Figure 2a,b show the von Neumann and CIM architecture to detect the pose. On von Neumann architecture, the sensed data, i.e., raw images, are stored on the memory and processed on the processing element, i.e., Raspberry Pi. However, on CIM architecture, the sensed data are stored and processed on the CIM.

This work uses Panasonic Grid-EYE thermal sensors, an 8×8 pixels infrared array sensor, to collect the thermal images of the subjects. The thermal images are calibrated to eliminate distortion and pre-processed to eliminate background noise on a low-power micro-processor and than upload to FPGA, a programmable integrated circuit. A pre-trained convolution neural network (CNN) model will also upload to the FPGA, which will read, write, and operate the CIM SRAM to conduct the convolution layers on CIM. The framework also implements activation functions on FPGA, such as max-pooling and leakyReLU. The results show that the accuracy of pose detection can be improved by at least 25%, compared to the method without enhancing the thermal image on CIM.

The remaining of this paper is organized as follows. Section 2 presents the background and related works, and Section 3 presents the architecture of the designed framework. Section 4 presents the designs and implementation for the framework, and Section 5 presents the evaluation results. Finally, Section 6 summarizes the work.

## 2. Background and Related Works

Digital and analog sensors have been developed to detect various events. In the last few decades, the sensing technologies have developed to detect semantic events, including presence of a human or a particular person, the presence of certain gases and temperatures, and changes in temperature or brightness. Computations are required to process raw sensing data and to obtain semantic information. This section presents the background of sensing technology and research related to this work.

### 2.1. Thermal Sensors

This work uses thermal images as the example of raw inputs to demonstrate and evaluate the framework of CIM-based smart sensors. Thermal sensors can have the readings for single point or array of points. The single point thermal sensors are often used to read the temperature on front-heads and single point of an object. On the other hand, the array thermal sensors are used to reconstruct of heat-map of an area and, hence, are used to detect abnormal temperatures in an area or a surface. The body temperature readings are widely used in public area to identify the person(s) having fevers. Chou et al. [7] developed the framework to measure gait velocity using connected low-resolution array thermal sensors.

Figure 3a,b show two thermal heat-map examples of an 8×8 pixels thermal image and an 120×160 pixels thermal image. Each pixel has a temperature reading. Specifically, the examples were collected from Panasonic Grid-EYE [8] and FLIR LEPTON [9] at the same time and location. Panasonic Grid-EYE thermal camera can output 8×8 pixels thermal data with 2.5 °C accuracy and 0.25 °C resolution between −20 °C and 80 °C at 10 frames per second. As shown in Figure 3a, it can observe the indoor area to recognize the daily activity but preserve privacy compared to surveillance camera and high-resolution thermal image sensors. This work uses ultra-low-resolution thermal sensors to collect the heat-map of the region of interest.

### 2.2. S-RAM CIM

Traditional sensors use either micro-processors or micro-controllers to process the collected data. Both of them use von Neumann computing architecture, which separates memory and processing units, for flexibility in last 40 years. There is no question that von Neumann architecture enables programmability. However, it also imposes significant overhead to move data between memory and processing units when the system has to process the collected data repeatedly in order to obtain the desired information. A deep neural network is one such computing model and performs poorly on von Neumann architecture. Several works [3,10,11,12] showed that a customized SRAM chip can be used to conduct multiply-accumulate (MAC) operations for convolutional networks.

This work uses 6T SRAM CIM designed by Liu and his colleagues [6]. It consists of 16 computation banks, where all banks share 64 7-bit DACs. Each bank consists of 16×64 SRAM cells. Each computing cycle triggers 1 column of each bank to conduct 16 matrix vector multiplies at the same time. Then, the output of all the columns will be summed up and output to ADC. Hence, the CIM chip can conduct multiplication and accumulation (MAC) in parallel. Equation (Equation 1) shows an example of MAC, which is the core computation operations of neural network algorithms:(1)$$\begin{array}{cc} Y_{i} & =round(\sum_{j=0}^{j<64}\left(X_{j}\times W_{j,i}\right)/64) \end{array}$$
where X is a 1×64 7-bit input array, and W is a 64×256 binary weight matrix.

Note that the results of MAC on analog CIMs do come along with errors because of the nonideality from ADC and DAC while reading or writing the SRAM. Olleta et al. shows the transfer curve between ideal values and real values of a 3-bit flash ADC with a voltage reference equal to 2 V in Figure 2 in [13]. As shown in that figure, the mapping from the ideal value to the real value is not linear and needs to be measured for characterizing the ADC under test. The nonideal transfer curve makes it difficult to use CIM as the processing elements. Hence, when using CIM as the processing element for neural network algorithms, the major challenge is how to characterize the transfer error and how to minimize the impact to the correctness of the computation results.

### 2.3. Quantization Aware Training/Post-Training Quantization

Analog CIM circuits also limit the number of bits for computation. The device designed by Chiu et al. [11] used 55 nm technology to design and has 2 to 8 bits as the input width, 2 to 8 bits as the weight width, and 7 to 19 bits as the output width. Similarly, the device designed by Hsu et al. [6] used 28 nm technology to design and has 7 bits as the input width, 1 to 2 bits as the weight width, and 5 to 7 bits as the output width. Compared to modern digital processors, these CIM SRAM designs have extreme limitations on data representation on inputs, weights, and outputs. Hence, the inputs and weights to be used on analog CIMs have to be quantized onto a specific number of bits in order to fit the circuit design. Quantization aware training usually has better accuracy, but takes time to retrain the model with specified precision. However, post-training quantization takes advantage of the pre-trained model, but the accuracy declines significantly when the specified precision (i.e., 7 bits) is much lower than the expected precision (i.e., 32 bits or 64 bits).

Figure 4a,b show the results of the post-training quantization SRCNN model and quantization-aware training SRCNN model [14,15,16]. By retraining the model instead of quantizing the pre-trained model, the peak signal-to-noise ratios (PSNR) of the super-resolution (SR) images increase from 21.5 dB to 22.9 dB. The images enhanced by the post-training model look after those suffered from applying a low-passed filter.

### 2.4. Pose Detection on SRCNN-Enhanced Thermal Images

Shermeyer and Van Etten [17] show that super-resolution can improve the accuracy of object detection on low-resolution images. Shih et al. [18,19] show that the accuracy of pose detection on a low-resolution thermal image can be improved by fusing the data from multiple sensors and an SRCNN [20] trained by an extra high-resolution thermal camera. They put four 8×8 resolution thermal sensors on the corner of a 10 cm × 10 cm plastic board and a 120 × 160 high-resolution thermal camera in the middle of the board to collect the data for training SRCNN. Compared with bi-cubic interpolation, SRCNN can enhance the PSNR from 8.3 dB to 19.5 dB, and the accuracy of the pose detection also increased from 77.93% to 89.30% on modern micro-processors.

## 3. System Architecture

Processing sensor data by Neural Network algorithm on von Neumann architecture microprocessors consume significant amount of resources, including CPU, GPU, storage space, and network bandwidth. The designed framework takes advantage of analog CIM SRAM to store and process-collected data on the sensors, rather than on the edge servers or remote cloud servers. Figure 5 shows the overall system architecture of CIM-based smart sensors for pose detection.

The framework consists of three major components: one low-power micro-processor, one FPGA, and one CIM SRAM. The sensor box shown on the left collects raw data for processing. The details of the sensor box will be illustrated later. A Raspberry Pi board downloads raw images from the sensors and pre-processes the image to eliminate noise and background. The FPGA is responsible for converting the received image onto the format for CIM and conducts the activation layer in the neural networks. The FPGA also writes the weights onto CIM SRAM and sequentially writes the input data into the DAC buffer. On the other hand, it retrieves the ADC outputs after the CIM finishes the computation. When the neural network algorithm completes on CIM, the FPGA will upload the feature maps to Raspberry Pi to reconstruct the image for SRCNN. While using the framework for pose detection, the FPGA can output the detection results to the display.

Figure 6 shows the thermal sensor box for collecting the infrared data. It consists of four Grid-EYE [8] sensors and one Lepton [9] camera. The Grid-EYE sensors have ultra-low resolution and are used for online pose detection; the Lepton thermal image sensor is used to collect high-resolution reference images for training. Auduino is responsible for transmitting raw thermal images to the external receiver. Raspberry Pi is responsible for formatting high-resolution thermal images for output. Thermal data are collected from a thermal sensor box, and the preprocessing is perfromed on Raspberry Pi. The preprocessing consists of fusing the data from Grid-EYEs into single image and crops to the target ROI. The details can be found at Shih et al. [18].

Conducting the computation by the circuits can have very high throughput and energy efficiency. Still, it is not easy to perform the computing other than addition, such as division or exponentiation. However, a convolutional neural network consists of convolutional layers and activation functions. This proposed framework integrates the FPGA and the CIM SRAM and designs a CIM-friendly CNN model which uses fewer channels, a smaller kernel size, and customizes the activation function. The goal of this framework is to be deployed on FPGA and support the CIM’s quantization and uncertainty computation.

## 4. CIM-Friendly Deep Neural Network Training and Inference

This section presents the process of CIM-friendly deep neural network training and inference. As an example, we designed a two-layer SRCNN without a reconstruction layer and a four-layer CNN pose-detection model to demonstrate the process.

The training and inference process was designed according to the system architecture shown in Figure 5. On the Raspberry Pi, it fuses the data from four sensors by bi-cubic interpolation and crops the image to 30×40 pixels resolution. After data preprocessing, Raspberry Pi will upload the low-resolution images to FPGA through UART. FPGA then writes the input data and weights to CIM SRAM and retrieves the result from ADC from CIM SRAM. Between two CIM layers, the normalization layer and LeakyReLU will be conducted on FPGA. When the SRCNN is conducted, the FPGA sends the feature map to Raspberry Pi to reconstruct the thermal image; when the pose detection is conducted, the FPGA outputs the detection results to display. To train a CIM-friendly CNN model, the process must take into account the errors on CIM SRAM. The errors include quantization error, non-ideality transfer error, and voltage bias errors.

### 4.1. Characterization of Non-Ideality Errors

The DAC bias-voltage and the ADC errors are the main causes for CIM non-ideality errors. This subsection elaborates how errors on CIM SRAM are characterized.

#### 4.1.1. Non-Ideality Transfer Errors Characterization

The non-ideality transfer errors are caused by the non-ideality transfers in ADC, which are essential in analog CIM SRAM to switch between digital and analog signals. To completely understand the characteristics of the CIM SRAM, the following three methods are used to collect the error-mapping tables:**Random Input**: in this method, the input are randomly selected in the range [−63,63], and the weights are also randomly selected between −1 and 1.**Color Image Data**: in this method, the inputs are color images and randomly selected from CIFAR-10 dataset, which is not the target dataset for training and testing.**Thermal Data**: in this method, the inputs are thermal images and randomly selected from thermal images collected by our sensor box. The main characteristics of thermal images are the input for the first layer are all positive.

Given the inputs from the above different distributions, the process collects the outputs and computes their means and standard deviations. For example, for a given input value, the ideal output of a given MAC operation is supposed to be *a*, and the mean of the collected output is *b*. All the output value *b* on CIM will be mapped to *a* to verify its correctness. Figure 7 shows the mapping from different inputs.

The blue line shows the ideal mapping, the red line shows the mapping for random input sources, the yellow line shows the mapping for inputs from the thermal image dataset, and the green line shows the mapping for the color image dataset. Although these four lines have similar trends, they do have different characteristics. The main difference between these two inputs are the scope of the values: the sampled inputs use thermal images where all values are positive, and the diversity in one convolutional window is small; the color images also have positive values but wider range; and the random inputs use the complete range [−63,63] as the sample space. Among three collected mappings, the random inputs lead to the closest to the golden mappings. On the other hand, the color image inputs lead to the least close to the golden mappings. The three different mappings will be used to train the networks and evaluate their testing accuracy.

#### 4.1.2. Errors Caused by DAC Bias-Voltage

Another cause of the non-ideality errors comes from the bias voltage on DAC. Figure 8 shows the difference on output values between high and low bias input voltages. When the bias-voltage is low, the CIM SRAM performs very similarly to the expected (gold) one, plotted by the gray line. However, it only makes use of the voltage range between −0.5 V and 0.5 V, which cannot fill the entire sampling range of the DAC and wastes the limited output precision. High bias-voltage can fully use the sampling range of the DAC, but the errors on both ends of the input ranges are unbearable. Selecting an appropriate bias-voltage for the user context can have a significant impact on the system.

### 4.2. Uncertainty-Aware Training

The training process of the networks to be executed on analog CIM SRAM needs to tolerate the errors characterized in the above subsection. This subsection describes the uncertainty aware training process.

The uncertainty aware training process has two types: **1-to-1 mapping** and **Gaussian mapping**. The first method uses the mean of the collected values for a given input value as the mapped value during the training process. Hence, it is called *1-to-1 mapping*. The advantage of this method is its low complexity by using a look-up table for the mapping. The second method uses the Gaussian distribution to map the output value during the training process. Hence, it is called *Gaussian mapping*. Figure 9 shows the five example distributions of the mappings. The orange bar shows the histogram of the collected outputs for an expected output, and the blue line shows the Gaussian distribution based on the mean and standard deviation of the collected outputs. As an example, Figure 9c shows the histogram and distribution of the collected output when the expected output is 0. The figure shows that the distribution greatly fit the histogram. Note that the distribution may not fit the histogram when the expected outputs are greater than 15 or less than −15. This is because the number of samples is smaller and, hence, the mean and standard deviation do not represent the distribution of the collected samples.

Based on the above observations, the Gaussian mapping method uses the distribution of each expected output value to map the output value during the training process. As a result, one output value may be mapped into different values before next operation to accommodate the errors caused by the non-ideality of the ADC circuits and bias-voltage.

### 4.3. Network Design for SRCNN on CIM

The SRCNN model used in this work includes three convolutional layers, and the first two layers and the activation function between them are computed on CIM SRAM and FPGA. The third layer of SRCNN, which is the reconstruction layer, will be conducted on a regular CPU or GPU.

The kernel size of the SRCNN model used in this paper is 7-1-5, and the channel size is 1-64-32-1. The first convolutional layer takes the 49 rows of the DAC buffer as the inputs and stores the weight on the 64-th columns, 4 columns for each bank. On each iteration, the CIM broadcasts the inputs and conducts a matrix array multiplication, and this operation will repeat four times for each input. A 30×40 image will update the inputs for 1200 times. The output feature map will apply a CIM-friendly normalization function and LeakyReLU before conducting the second convolutional layer. The second convolutional layer uses a 1×1 kernel because the input channel number of the second layer is 64 and fits the width of CIM SRAM.

### 4.4. Network Design for Pose Classification on CIM

Figure 10 shows the example thermal images which are used in this work for pose classification. The camera is set on the ceiling above the bed. The user might be sitting, lying down, or absent. Each class has 400 images for training, 100 images for testing, and 100 images for verification.

The pose classification model in this work has two types: the first one uses the low-resolution thermal image as the input, and the second one uses the feature map of a thermal image after applying the first two layers of SRCNN shown in Figure 11. The direct method has a faster inference time and fewer parameters to train. However, the SRCNN method might have a better outcome but need more time to adjust all parameters, and a deeper neural network might suffer from the uncertainty of CIM computation.

This work also uses a customized normalized function, an activation function, and a max-pooling after each convolutional layer.

Customized Normalized Function:Since the output resolution of CIM computation is very small, most of the output values are less than 20. Hence, if the outputs were not enlarged after two or three convolutional layers, all outputs will become zero. The proposed normalized function adjusts the offset of output values to reduce the non-ideality error and enlarge the outputs by multiplying the power of two.For a CIM SRAM with 7-bit input and 7-bit output precision, the range output will be much smaller than input for most of the time. Hence, we only need to focus on the uncertainty of small values to fix the error. While training a CIM-friendly CNN model, the model will search for a scale factor and an offset such as the γ and β in Batch Normalization to maximize the accuracy. The convolutional function will become similar to the following equation:
(2)$$\gamma \times \sum_{i=0}^{n}I_{i}\times W_{i}+\beta$$
where *n* is the number of chunks, $I_{i}$ is the inputs, $W_{i}$ is the weight, γ is the scaling factor, and β is the offset.Activation Function:This work uses LeakyReLU as the activation function because it is easy to implement on FPGA, and instead of ReLU function. At the same time, the LeakyReLU can still preserve the information from a negative value. Since the CIM outputs are integers and the range is very small, we use 0.5 as the negative slope of the LeakyReLU.

The network accepts two different inputs for pose detection: **thermal images** and **Feature Map from SRCNN**.

Thermal Image Inputs:This pose-detection network uses a four-layer CNN model with three convolutional layers and one fully connected layer to process the thermal images as inputs. The kernel size of the convolutional layers is 3×3, and the size of the Max pooling layer is 2×2.Feature Map from SRCNN:Many previous works show that the accuracy of object detection gains benefits from SRCNN. However, due to the property of CIM computation, it can only conduct the CNN with quantized weights, and the quality of most images is even worse than the image only applying bicubic interpolation. Therefore, we design a three-layer SRCNN model trained by Grid-EYE and Lepton images. After the model is trained, the framework only use the feature map output from the second layer as shown at the upper half of Figure 11. The feature map will be used to train the four-layer pose classification model.

The networks are trained with 150 epochs, where the batch size is 100. The optimizing function is stochastic gradient descent with a learning rate of 0.001 and a momentum factor 0.9. The negative slope of LeakyRelu is 0.5.

## 5. Performance Evaluation

This section presents CIM simulation results and on-chip inference results. The neural networks are trained on an uncertainty-aware and quantized-aware SRCNN model in a CIM-simulated environment. The evaluation compares the difference between simulated and inference results.

### 5.1. Experimental Setting and CIM-Simulated Environment

The performance of the proposed framework is evaluated on both physical devices and simulated environments in order to study the effectiveness of the uncertainty-aware training process. Figure 12 shows the experimental setting for CIM SRAM and FPGA. The figure shows the CIM SRAM chip, FPGA, level shifter, and personal computer. The computer is used to emulate the sensor box and Raspberry Pi in order to obtain consistent and fair results via sending thermal images and receiving the results. The level shifter is deployed between the FPGA and the CIM SRAM chip to synchronize the voltage between the two devices.

To simulate the CIM environment, we rewrite the convolution functions in Pytorch. The simulated environment splits the channels into several chunks according to the kernel size. Each chunk has less than or equal to 64 inputs. For example, a 3×3 kernel will be split into 4 channels in a chunk. After conducting the convolution chunk by chunk, the output will apply the method elaborated in Section 4.2 to simulate the CIM errors.

### 5.2. Parameters for Activation Functions and Normalization Functions

This subsection presents the parameter configuration for activation function and normalization functions, which are conducted on FPGA.

Activation Function: ReLU vs. LeakyReLU

Considering the complexity of hardware implementation, our model uses ReLU as the activation function in the first design. However, due to the uncertainty error of CIM computation, the output is smaller than expected and makes the most of outputs become negative and zeroed by ReLU. In our testing data, the most output on an ideal CIM will be −10 to +10, but in the simulated environment, it will be −17 to +3.

On a modern GPU, the accuracy is 98.7% with binary weight and 99.5% with floating weight. In the mean time, the detection accuracy of a three classes classification on an ideal CIM chip is 92.9% on average due to the limited computation accuracy and capability. Table 1 shows the accuracy of the simulated CIM model using ReLU or LeakyReLU with different negative slopes. While using normal ReLU as an activation function, the accuracy of the simulated CIM becomes the same as a random model. However, if we replace the ReLU with LeakyReLU with a 0.25 negative slope, the accuracy can be restored to 51.7%.

Customized Normalization Function

While training a CIM-friendly CNN model, the model will compute a scale factor and a shift offset as the γ and β in Batch Normalization to better use the limited bits for computation. Consequently, the convolutional function becomes the following equation: (3)$$\gamma \times \sum_{i=0}^{n}I_{i}\times W_{i}+\beta$$
where *n* is the number of chunks, $I_{i}$ is the inputs, $W_{i}$ is the weight, γ is the scaling factor, and β is the offset. The accuracy can be improved to 91.5% shown in Table 2 when γ is 4 and β is 2. Consequently, the output values of each layer will not be biased to certain ranges.

### 5.3. SRCNN on Pose Detection

To evaluate the effectiveness of SRCNN on thermal images, we use a dataset with five different poses, as shown in Figure 13, to evaluate the framework. Table 3 compares the pose detection accuracy when the SRCNN is conducted to when it is not. The top row shows the detection accuracy when the thermal images are not processed by SRCNN. In other words, layer 1 and 2 shown in Figure 11 are skipped. The bottom shows the detection accuracy when all the layers shown in Figure 11 are conducted. Three columns in Table 3 show the detection accuracy for the networks’ simulated CIM-based platform in order to evaluate the effectiveness of SRCNN. The first column presents the results of simulating the binary CIM with non-ideal errors, the second column presents the results of simulating the binary CIM without non-ideal errors, and the third column presents the results of floating-point platforms. The normalization parameters γ of layers 3, 4, and 5 of the CNN networks for 2 simulated CIM platforms are listed in the top row.

The detection accuracies of the pose-detection networks, from layers 3 to 6, shown in Figure 11, is 52.4% and 70.6% for binary weight CNN model on non-linear and ideal CIM-based platforms, respectively. The non-linearity transfer errors do not reduce the detection accuracy. When the floating point weight model is applied, the detection accuracy becomes 84.4%. After applying the first two layers of the SRCNN model, from layer 1 to 2 shown in Figure 11, the detection accuracy was enhanced to 72.2% and 82.5% for non-ideal and ideal CIM-based platforms. Again, when the floating point weight model is applied, the detection accuracy becomes 92.1%. The above results show that applying SRCNN does enhance the detection accuracy at all three computing configurations.

### 5.4. Transfer Curve Models

The non-ideal errors on the CIM-based platforms are calibrated by the transfer curves during the computation. However, the method to build the transfer curve can have impacts on the computation results. This subsection evaluates the computation results for different transfer curve models. Two models are evaluated in this subsection. The first model uses random values between [−63,63] to build the transfer curve model; the second model uses the random values from sample datasets to build the transfer curve model.

Table 4 compares the SRCNN results for two transfer curve models. Column (a) shows two sample input images, and Column (b) shows the results from simulated ideal CIM-based platforms where there are no non-ideal errors. Column (c) and (d) show the results using the weight which is trained by the error model from [−63,63] and the sampled dataset. The SSIMs for the 2 models are 0.45 and 0.65: there are 44% improvement when the transfer curve model is built upon the sample data.

The structural similarity index measure (SSIM) index compares the similarity of two images. The resultant SSIM index is a decimal value between 0 and 1. A value of SSIM = 1 is only reachable in the case of 2 identical sets of data and therefore indicates perfect structural similarity. A value of SSIM=0 indicates no structural similarity.

### 5.5. Training with Gaussian Noise

Given the transfer curve model, the framework may use either 1-to-1 mapping or Gaussian mapping, as shown in Section 4.2. The subsection evaluates the effectiveness of these two mapping methods.

Table 5 shows image results while conducting the first layer of SRCNN on CIM using these two mapping methods. Column (a) shows the results from simulated ideal CIM, Column (b) shows the results while using 1-to-1 mapping method, and Column (c) shows the results for using Gaussian mapping method. The SSIM of the examples on first row increases from 0.70 to 0.81, and that on the second row increases from 0.61 to 0.76. The result shows that using the Gaussian mapping method to emulate the non-ideal errors during training can better tolerate the uncertainty on CIM devices.

### 5.6. Energy Consumption

The energy consumption is evaluated on conducting the convolution networks, which is the most intensive part of the entire computation flow, on the ARM-based platform and the CIM-based platform. The energy consumption is measured on both platforms when conducting the convolution networks computation because all the other computations are conducted on the same computing devices and have no impact on the difference in energy consumption. Figure 14 illustrates the setting of energy consumption measurement. The ARM-based platform, which is a Raspberry Pi device and shown as the left red dashed rectangle in Figure 14, is connected to a USB power meter. The CIM-based platform, shown as the right red dashed rectangle in Figure 14 is connected to a Tektronix Keithley DMM7510 (https://www.tek.com/en/products/keithley/digital-multimeter/dmm7510 last accessed on 10 January 2022) to measure the energy consumption.

The computation workload for the measurement is conducting the Layer 1 and Layer 2 of the pose detection flow, shown in Figure 11, for 100 thermal images. On the ARM-based platform, the energy consumed for Layer 1 and Layer 2 are 14.55 mJ and 9.9 mJ, respectively. On the CIM-based platform, the energy consumed for both Layer 1 and Layer 2 are 0.00029 mJ. The reduction in energy consumption ranges from 50,000 and 34,100 times.

## 6. Summary

This work proposes a framework to take advantage of CIM SRAM for developing smart sensors and evaluates its performance. CIM SRAM can conduct matrix multiplication with low energy consumption and high throughput. However, it only supports integer input and output, making the rounding error a significant challenge while training a learning model. Moreover, the uncertainty error is another non-negligible challenge at inference. This work proposes a systematic method to measure the error model of CIM SRAM and a normalized function to overcome the rounding and uncertainty error. With proper parameters and the LeakyReLU function, the accuracy of the three classes classification can increase from 33.3% to 91.5%. This work also studies the impact of executing SRCNN on CIM SRAM. The experiments show that running SRCNN on CIM SRAM may not increase the quality of images, but using the feature maps from SRCNN instead of raw low-resolution images can help the classification layers distinguish the poses.

## Figures and Tables

**Figure 1 sensors-22-03491-f001:**
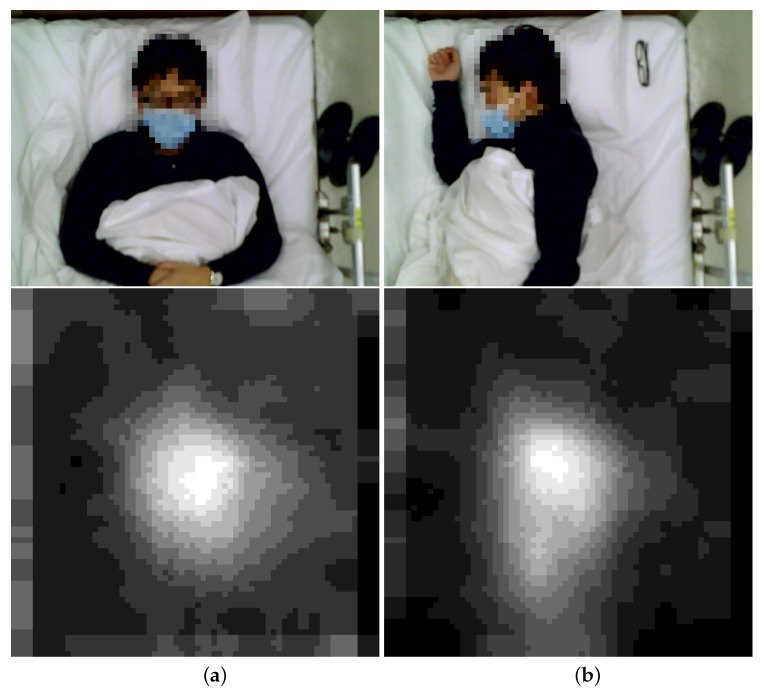
Color Photos and Thermal Images at Okayama Hospital, Japan: (**a**) Laydown, (**b**) Turn to the right.

**Figure 2 sensors-22-03491-f002:**
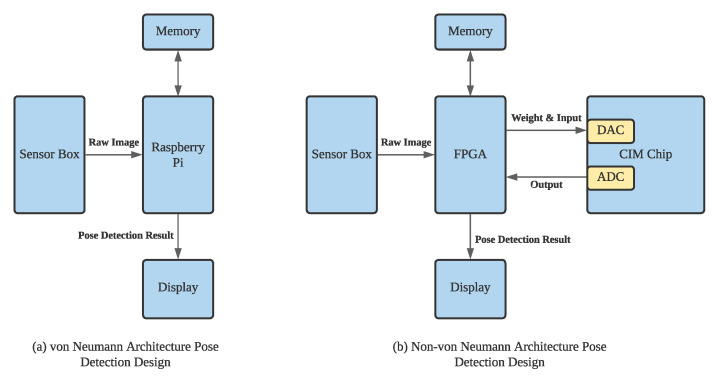
The von Neumann and CIM Architecture for Thermal Image Processing.

**Figure 3 sensors-22-03491-f003:**
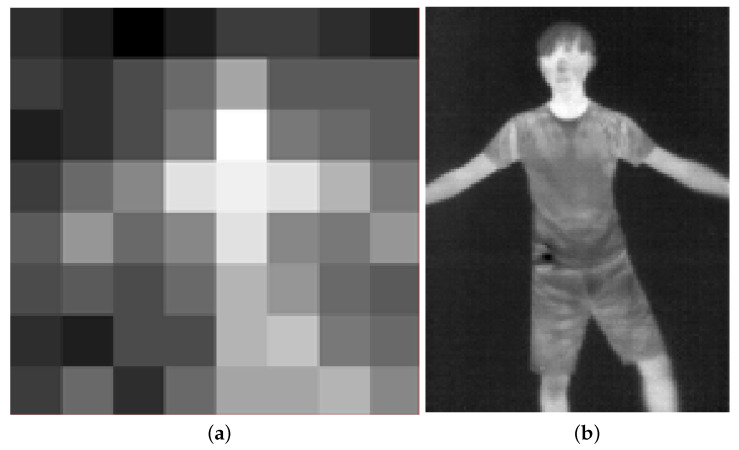
Examples of Thermal Heat-Map Images: (**a**) Low-Resolution Thermal Image; (**b**) High-Resolution Thermal Image.

**Figure 4 sensors-22-03491-f004:**
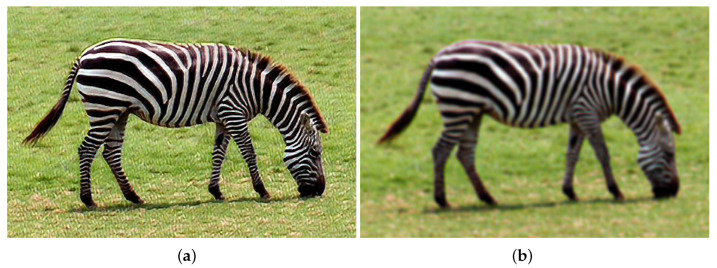
Quantization SRCNN: (**a**) Post-Training Quantization; (**b**) Quantization Aware Training.

**Figure 5 sensors-22-03491-f005:**
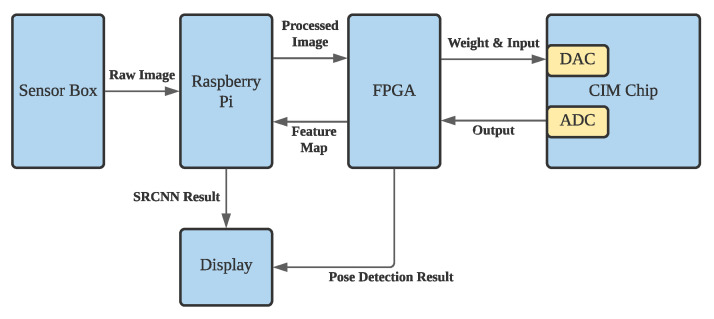
Overall System Architecture of CIM-Based Smart Sensor.

**Figure 6 sensors-22-03491-f006:**
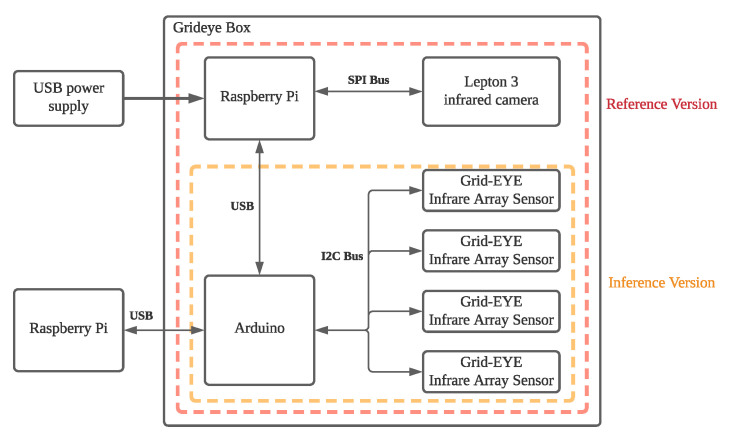
Thermal Box Design.

**Figure 7 sensors-22-03491-f007:**
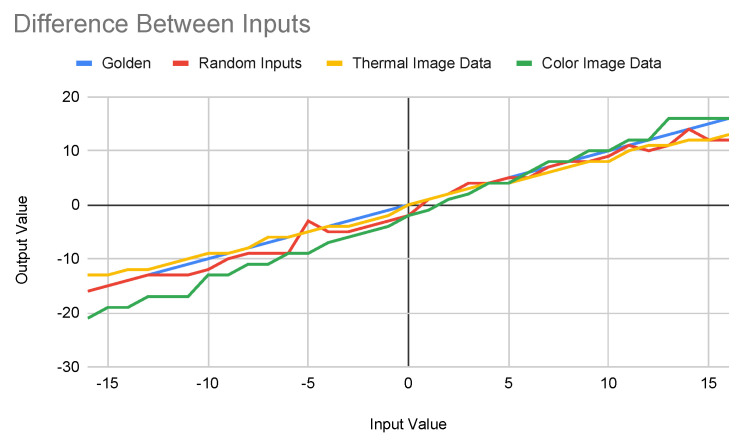
Mapping on Different Input Sources.

**Figure 8 sensors-22-03491-f008:**
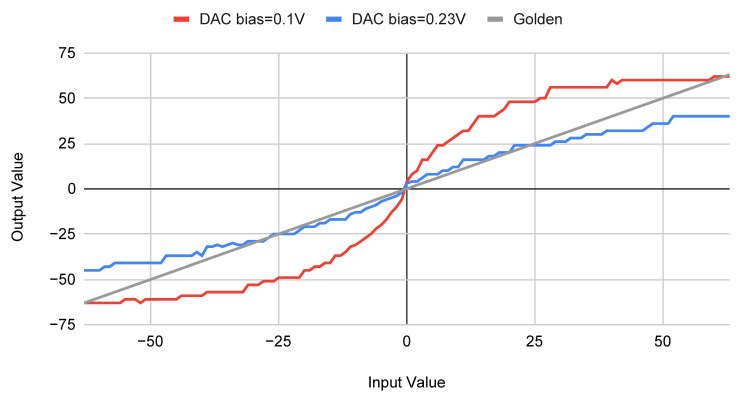
Difference Between Bias-Voltage.

**Figure 9 sensors-22-03491-f009:**
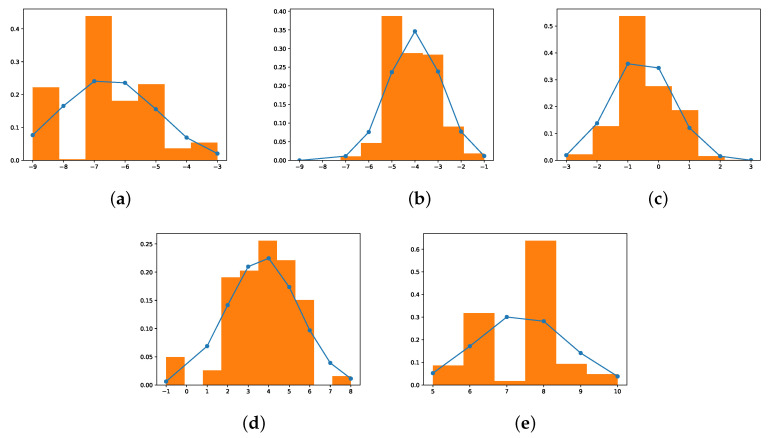
Error Distribution: (**a**) Expected = −6; (**b**) Expected = −3; (**c**) Expected = 0; (**d**) Expected = 3; (**e**) Expected = 6.

**Figure 10 sensors-22-03491-f010:**
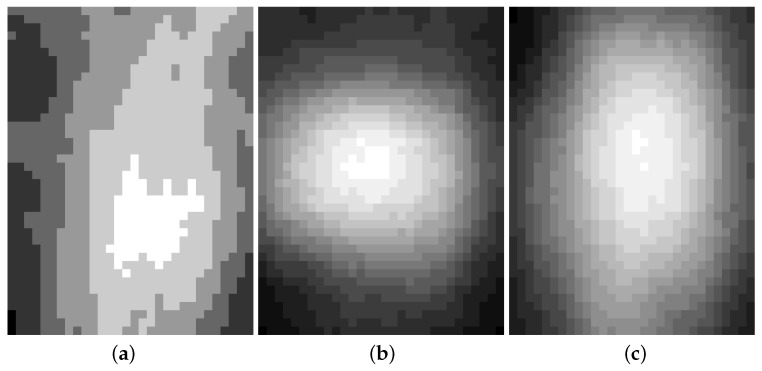
Example Data for Pose Detection: (**a**) Empty; (**b**) Sitting; (**c**) Laying.

**Figure 11 sensors-22-03491-f011:**
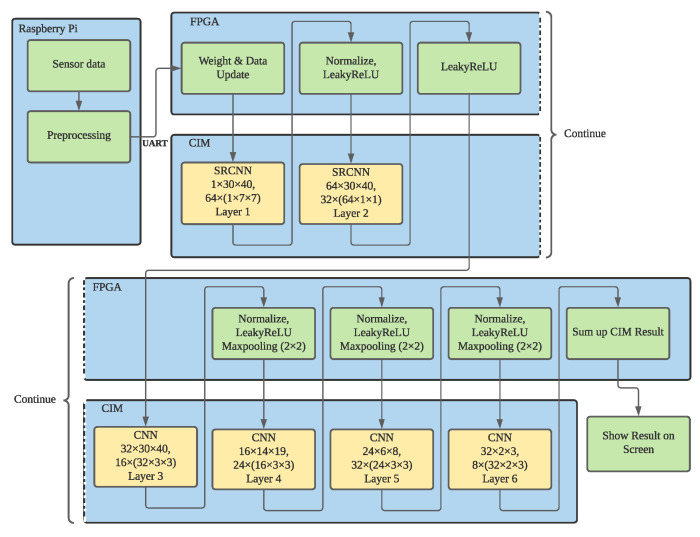
Network Architecture for Pose Detection.

**Figure 12 sensors-22-03491-f012:**
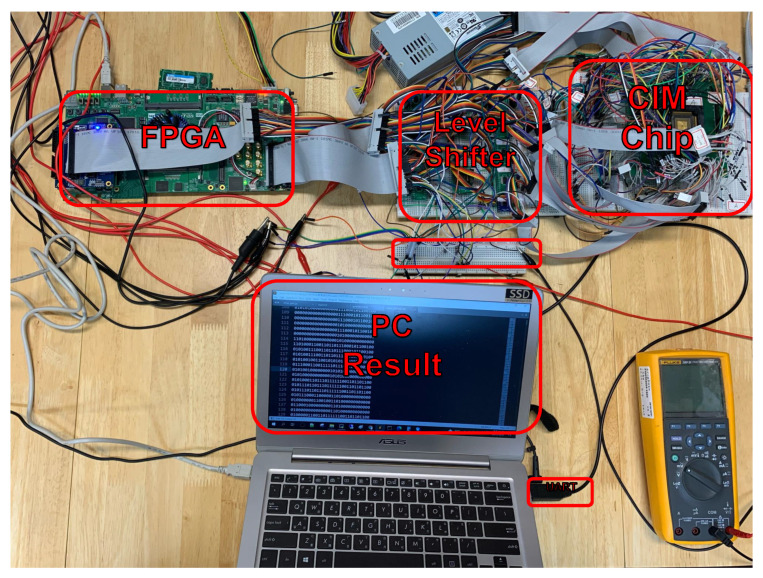
Experimental Setting for CIM SRAM and FPGA.

**Figure 13 sensors-22-03491-f013:**
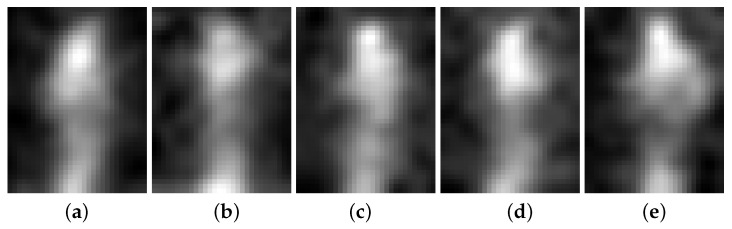
Input Images for Pose Detection: (**a**) Standing; (**b**) Raising hand; (**c**) Arms on hips; (**d**) Crossing hands; (**e**) Hands on hips.

**Figure 14 sensors-22-03491-f014:**
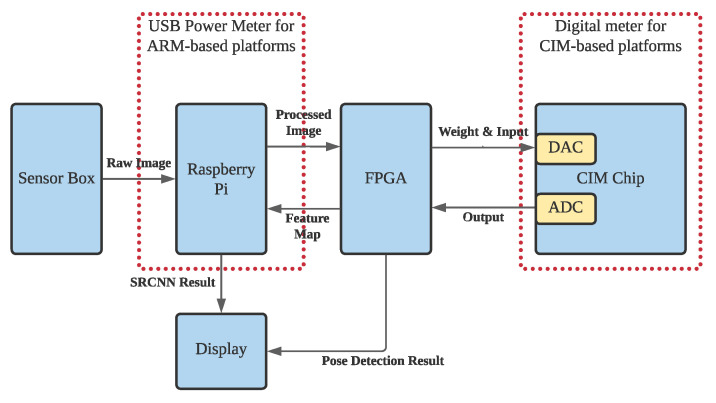
Energy Consumption Measurement for ARM- and CIM-based platforms.

**Table 1 sensors-22-03491-t001:** The accuracy of three classes classification with different activation functions.

ReLU	LeakyReLU(0.5)	LeakyReLU(0.25)
33.3%	34.6%	51.7%

**Table 2 sensors-22-03491-t002:** The accuracy of different normalized parameters with LeakyReLU(0.5).

γ=4,β=0	γ=4,β=1	γ=4,β=2	γ=8,β=0	γ=8,β=1	γ=8,β=2
61.0%	77.9%	**91.5%**	66.9%	81.7%	66.0%

**Table 3 sensors-22-03491-t003:** The accuracy of five classes classification with SRCNN.

	**On Non-Ideal CIM** γ=(4,4,2)	**On Ideal CIM ** γ=(8,4,2)	**Floating CNN**
Accuracy without SRCNN	52.4%	70.6%	84.4%
	**Feature Map of SRCNN On Non-Ideal CIM **	**Feature Map of SRCNN On Ideal CIM**	**Feature Map of SRCNN and Floating CNN**
Accuracy enhanced by SRCNN	72.2%	82.5%	92.1%

**Table 4 sensors-22-03491-t004:** SRCNN results under different error models.

Input Images	Simulated SRCNN	Error Model by Random Inputs	Error Model by Sampled Dataset
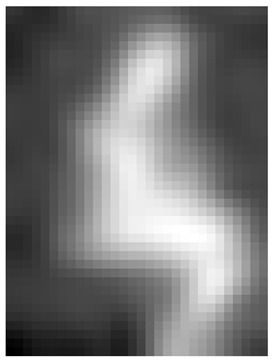	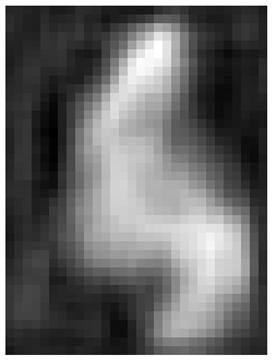	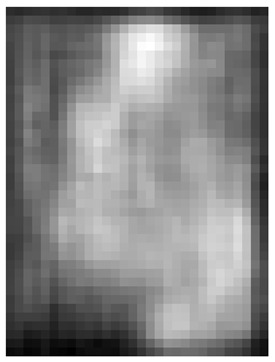	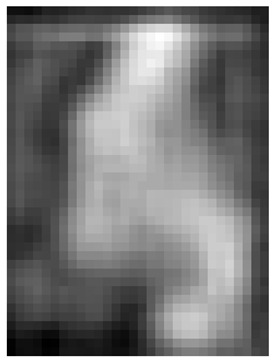
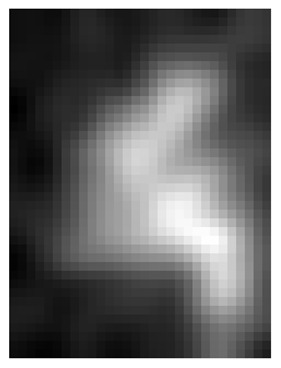	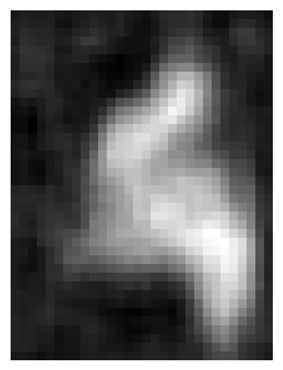	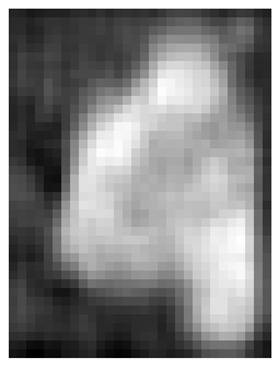	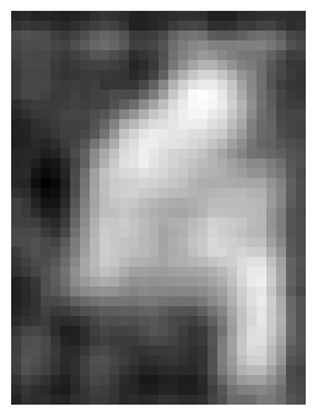
		SSIM = 0.45	SSIM = 0.65
(a)	(b)	(c)	(d)

**Table 5 sensors-22-03491-t005:** SRCNN Results when conducting only the first layer on CIM.

Simulated Ideal CIM	Trained with 1-to-1 Mapping	Trained with Gaussian Mapping
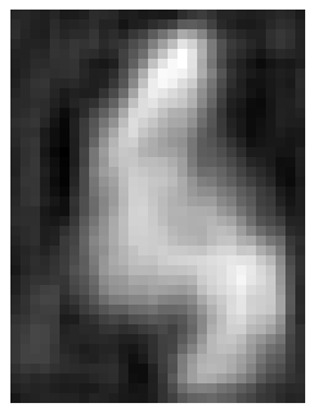	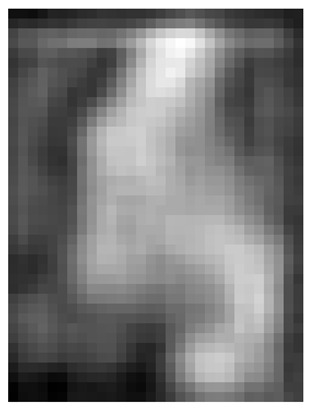	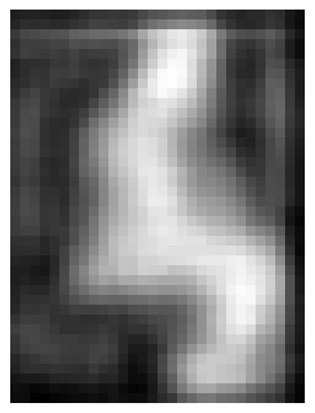
	SSIM = 0.70	SSIM = 0.81
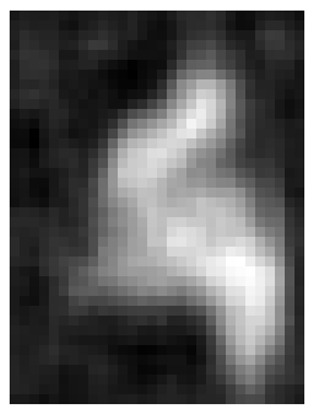	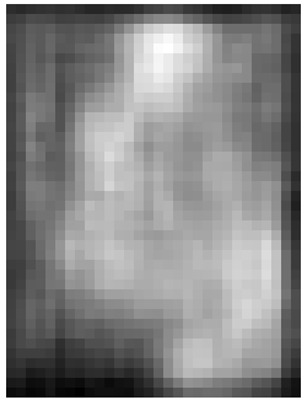	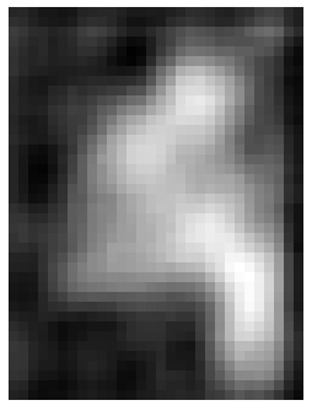
	SSIM = 0.61	SSIM = 0.76
(a)	(b)	(c)

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
