# Peer review of "CIM-Based Smart Pose Detection Sensors"

_sensors, 2022, doi:10.3390/s22093491_

Round 1

Reviewer 1 Report

This paper proposed the CIM-based sensing architecture for human pose recognition. However, I also believe it requires some modifications to deserve publication in the SENSORS. I will try to point out some of my concerns in the following, hoping you would consider these only as suggestions to improve the paper.

1. This paper proposed a new architecture of the human pose detection sensors' digital processing elements using the computing-in-memory (CIM) which is published in [3]. The authors claim that this new architecture reduces the computation energy and improves the recognition accuracy compared to the conventional microprocessors. In Fig. 1, the block diagram of the proposed pose detection system using the CIM processing element. If the figure of the pose detection system uses conventional von Neuman micro-processors, it will be helpful for readers to understand the structural advance of the proposed system. 
Please add a block diagram of the pose detection system using conventional microprocessors.

2. The most important advantage of this paper is the reduction of energy consumption. In Section 1, the authors claimed that the energy consumption can be reduced 50000 times by using proposed architecture with CIM (0.00029mJ) than when using convenntonal architecture with Raspberry PI 3 (14.55mJ). 
Please explain which blocks in Fig. 1 consumes the energy (0.00029mJ). Is it the energy consumption of the entire system in Figure 1? Or is it the energy consumption of the CIM block (including DAC, ADC) in Figure 1? 

3. Please add the measured energy consumption of the entire system in Figure 1 and Figure 11. Also, please add the measured energy consumption of each blocks in Figure 1 and Figure 11.
Through this, it is possible to clearly understand the proportion of digital processing block (CIM) in the total energy.

4. In Section 2.4, it was explained that PSNR and accuracy were improved due to the use of SRCNN, but the discussion on data was omitted.
In Section 2.4, please discuss why SRCNN can increase PSNR and accuracy.

5. In Section 4.2 "Errors Caused by ADC Bias-Voltage", the effect of ADC bias voltage is explained using Figure 7. However, Fig. 7 is a graph of the effect of DAC bias voltage, not the ADC. 

6. There is no detailed description of the parameters γ and β used in Section 5.2 “Customized Normalization Function” section. Please add a detailed description of the parameter or add a related reference. 

7. Please add full text for abbreviations in the manuscript such as SRCNN, CNN, PSNR, and SR.

8. In Figure 10, because there are two FPGAs and two CIM boxes, it can be understood that two FPGAs and two CIMs are physically used. 

Author Response

  1. In Figure 2, illustrates the difference between von Neumann and CIM architecture for pose detection. On von Neumann architecture, the sensed data, i.e., raw images, are stored on the memory and processed on the processing element, i.e., Raspberry Pi. However, on CIM architecture, the sensed data are stored and processed on the CIM.
  2. We revise section 5.6 and Figure14 for energy consumption. In Figure 13, how the energy consumptions are measured for conventional and CIM-based platforms. The ARM-based platform, which is a Raspberry Pi, is connected to a USB power meter. The CIM-based platform is connected to a Tektronix Keithley DMM7510

  3. We revise section 5.6 and Figure14 for energy consumption. In Figure 13, how the energy consumptions are measured for conventional and CIM-based platforms. The ARM-based platform, which is a Raspberry Pi, is connected to a USB power meter. The CIM-based platform is connected to a Tektronix Keithley DMM7510

  4. In section 2.4, we discuss the PSNR and the classification accuracy can be improved by SRCNN. Compared with bi-cubic interpolation, SRCNN can enhance the PSNR from 8.3dB to 19.5dB, and the accuracy of pose detection also increased from 77.93 to $89.30% on modern micro-processors. Section 5.3 also presents the quality improvement on using SRCNN. The detection accuracy was enhanced to 72.2% and 82.5% for non-ideal and ideal CIM-based platforms.
  5. Fixed this typo.
  6. We add the detail in Section 4.4
  7. Fixed this problem.
  8. We redraw the figure to make it easier to understand.

Reviewer 2 Report

This paper proposes  a framework to take advantage of CIMSRAM for developing smart sensors and evaluates its performance. The accuracy seems reasonable, while data used is not clear. Some issues as follows:

  1. Acronyms usage - Each acronym should be defined at first and  used repeatedly. SCNN is used without definition (super resolution cnn?). CIM is defined multiple times.
  2. In Section 5.2, it is not clear how many datasets for five different poses are used for evaluation. Which dataset did authors use? Data statistics (types, resolution, real or synthesized) for training is not clear either.
  3. Training settings, such as learning rate, epochs, batch size are not clearly described.
  4. Although the accuracy of this method is acceptable, but the the authors do not compare with other similar methods. Please provide some comparison experiments to prove the effectiveness of this method.
  5. Abstract: The evaluation results show that the accuracy of pose recognition can be improved by more than 25% when the energy consumption of executing one convolution layer is only 50, 000 times of digital sensing system . However, no evaluation results in Section 5 is related to energy consumption and time consumption. Please add these evaluation results in detail.
  6. There are several typos and figures in the manuscript that require rephrasing/revision. For example P8 L246 : grange, P13, Fig 13 captions and arrangement.

Author Response

  1. Remove the unnecessary definition.
  2. We use different datasets to evaluate our method. Without SRCNN, we use the dataset with three classes. With SRCNN, we use the dataset with five classes.
  3. We added the detail in section 4.4. The networks are trained with 150 epochs where batch size is 100. The optimizing function is stochastic gradient descent with learning rate 0.001 and momentum factor 0.9.  The negative slope of LeakyRelu is 0.5.
  4. In Section 5.2, the comparison to modern GPU and floating point network are added, shown below. On a modern GPU, the accuracy is 98.7% with binary weight and 99.5% with floating weight. In the mean time, the detection accuracy of a three classes classification on an ideal CIM chip is 92.9

  5. We revise section 5.6 and Figure14 for energy consumption. In Figure 13, how the energy consumptions are measured for conventional and CIM-based platforms. The ARM-based platform, which is a Raspberry Pi, is connected to a USB power meter. The CIM-based platform is connected to a Tektronix Keithley DMM7510
  6.  For the 6th suggestion, we fixed the typos as the suggestion.

Round 2

Reviewer 1 Report

Authors explained most of the reviewer's doubts and questions, and the revised manuscript had been improved according to the comments.

Reviewer 2 Report

I examined the response letter and found that they responded to all my comments.